# Effects of cochlear implantation on gait performance in adults with hearing impairment: A systematic review

**Bahaa Rafoul[1,2], Roy Tzemah-Shahar[1], Anat V. Lubetzky[3], Mauricio Cohen-Vaizer[2], Hanin Karawani** ![ORCID][1,4*‡], **Maayan Agmon[1‡]**

1 Faculty of Social Welfare and Health Sciences, University of Haifa, Haifa, Israel, 2 Rambam Health Care Campus, Haifa, Israel, 3 Department of Physical Therapy, Steinhardt School of Culture, Education and Human Development, New York University, New York, NY, United States of America, 4 Cluster of Excellence Hearing4All, University of Oldenburg, Germany

‡ Equal senior contribution.
* hkarawani@staff.haifa.ac.il

## Abstract

### Background

Previous systematic reviews evaluated the effect of hearing interventions on static and dynamic stability and found several positive effects of hearing interventions. Despite numerous reviews on hearing interventions and balance, the impact of cochlear implantation on gait and fall risk remains unclear.

### Objective

This systematic review examines the effects of cochlear implantation on gait performance in adults with hearing loss.

### Methods

A comprehensive literature search was conducted in PubMed, Web of Science, and Scopus, using the Preferred Reporting Items for Systematic Reviews and Meta-Analyses (PRISMA) guidelines. The PEDro scale assessed the methodological quality, risk of bias, and study design of included articles.

### Results

Seven studies met the inclusion criteria. Five focused solely on cochlear implantation, while two included both cochlear implants (CIs) and hearing aids. Methodological inconsistencies were evident in measurement approaches and follow-up durations, leading to variable outcomes. Short-term follow-up post-implantation showed no improvement or even worsened gait outcomes. However, a longer follow-up of three months post-implantation indicated partial improvements in specific gait measures like Tandem Walk speed, though not in comfortable walking speed. Cross-sectional studies comparing on-off CI conditions revealed no significant differences in gait outcomes.

**Data availability statement:** All relevant data are within the manuscript and its Supporting Information files.

**Funding:** This work was partially supported by the Faculty of Social Welfare and Health Sciences grant (awarded to HK and MA], Rambam and Faculty of Social Welfare and Health Sciences dual grant [awarded to BR, MCV, HK, MA]; The MAOF Research Grant Program for Research in Nursing at the Rambam health care campus [awarded to BR].

**Competing interests:** The authors have declared that no competing interests exist.

## Conclusions

Improvements in gait due to cochlear implantation require at least three months to manifest. The variability in study methodologies complicates understanding the full impact of cochlear implantation on gait. Given that only seven, methodologically inconsistent articles were found, it is necessary to conduct additional research to understand the relationship between hearing, gait and fall risk and to specifically include longer post-CI monitoring periods.

## Introduction

Age-related hearing loss (ARHL) is a gradual phenomenon resulting from the aging process's collective effects on the auditory system; defined as a progressive, bilateral, symmetrical, age-related sensorineural hearing loss that is most pronounced at higher frequencies [1]. Strikingly, hearing loss is associated with 2.39 times increased risk of falling among older adults compared to those with normal hearing [2].

Normal Gait requires several functional balance systems and postural control [3] and is important for healthy daily living. Gait quality can be measured by walking features such as speed and variability between steps [4]. Slower walking speed is associated with a decline in physical and mental functional capacity [5], falls [6], poor health, and decreased survival [5].

Hearing impairment is associated with slower gait speed, reduced walking endurance [7–9], lower physical performance [10], walking difficulties, poor standing balance, and overall reduced mobility [11]. Difficulties in walking and the prevalence of falls – both related to balance – have been shown to increase with the severity of hearing loss [7,12]. Therefore, it is important to understand the possible mechanisms that explain the link between hearing, gait and balance during standing or walking.

When considering sensory mechanisms, auditory sensory input is an essential factor influencing postural control [13] for balance and gait. Reduced audibility negatively affects sensory integration for postural control and balance performance [14,15]. In addition, physiological and anatomical mechanisms (neural, vestibular, genetic, and vascular) refer to the anatomical proximity between the cochlea and the vestibular system, exposing both to infections and ototoxic medications that could endanger both systems [2,16], affecting both hearing and balance control at the same time. Hearing loss affects allocation of attentional and cognitive resources that are typically reserved for balance; this leads to a reduced ability to divide attention and thus reduced balance control [17–19]. Advances in behavioral research have revealed that reduced mobility leads to limitations in spatial awareness as well as other psychosocial challenges [2,14,20], such as depression [21], loneliness, and social isolation [20]. Individuals with hearing loss are also prone to accelerated cognitive decline [22,23] which influences their balance [3,17], eventually increasing a person's risk of falling [24]. Therefore, to overcome these associated functional declines, hearing interventions are essential.

Hearing interventions include hearing aids and/or cochlear implants (CIs), offered according to the severity of the hearing loss [25]. Studies have shown that neural deprivation induced by reduced audio-sensory input can result in alterations in the central auditory system [26,27], but the use of hearing aids in cases of mild-to-moderate hearing loss can positively influence cognitive function [28] as well as cortical function and structure [28,29]. Furthermore, restoring auditory processing activity through CIs may reverse central changes related to severe-to-profound hearing losses [30]. Other research has also uncovered that the use of CIs can decrease levels of depression, loneliness and improve quality of life [31].

Previous systematic reviews in adults evaluated the effect of hearing interventions on static and dynamic stability using posturography and found several positive effects of hearing interventions, such as improved ability to maintain posture for more extended periods and reduced body sway and reactive balance [31–33]. In line with this, previous research also shows that children with sensorineural hearing loss face significant challenges in balance, showing greater postural sway and instability [34–36], which is further affected by the severity of hearing loss [37], affecting both static postural control and dynamic balance during activities like walking, where sensory input integration is essential [3].

Interventions using auditory stimulations, especially CIs, have shown a potential to improve postural stability and balance, reduce postural sway, and positively affect dynamic motor skills like gait in children [38–41].

Among adults, a growing number of studies have investigated the effect of CIs on gait performance [42–46]. Yet, previous reviews have focused on the effect of CIs on static and dynamic balance, overlooking gait outcomes. Because gait is essential for daily activities, this review aimed to systematically analyze the literature that investigated the effects of CIs on gait among individuals with hearing loss.

## Materials and methods

This systematic review followed the recommended guidelines of Preferred Reporting Items for Systematic Reviews and Meta-Analyses (PRISMA) [47]. PRISMA statement is listed in S1. The review protocol described here was registered with PROSPERO (registration number CRD42022315866) on June 27th, 2022.

### Inclusion and exclusion criteria

Reviewed studies met the following eligibility criteria: Prospective interventional studies or cross-sectional studies with aided and unaided conditions, written in English, with human subjects 18 years or older, and hearing loss diagnosed with audiometric testing; hearing intervention by CI (unilateral and or bilateral); gait measured by objective measurements (e.g., speed, stride time/length variability, or by stopwatch); subjects were able to walk independently, with or without an assistive device. Exclusion criteria: Reviews, meta-analyses, commentaries, and opinion articles; study sample with neurological diseases, acute heart disease, and mental health disorders.

### Search strategy

We systematically searched the following databases: PubMed, Scopus, and Web of Science, with a last search on January 8th, 2024. We used the following search terms for all databases: (deafness OR presbycusis OR hearing loss OR hearing dis* OR hearing impair* OR auditory impair* OR auditory dis*) AND (walking OR gait OR dyn* balance OR mobility OR postural control OR posture) AND (Cochlear implant* OR hearing aid* OR auditory rehab* OR hearing rehab*).

### Article selection

All results of our search strategy were first screened for relevance by the first author (BR) in terms of title and abstract. The papers that were found relevant underwent full evaluation for eligibility by BR and RTS independently and independent results from the two investigators were compared. In one case, when one of the investigators was uncertain, two additional authors, HK and MA, reviewed the article, and all reviewers re-evaluated the full article so a consensus could be reached. Description of all the 2076 records can be found in S2.

### Data extraction

We extracted the relevant data from each of the included studies, namely the study's aim, sample size, design, population, inclusion and exclusion criteria, hearing status and relevant metrics, follow-up duration, gait tasks and performance, vestibular measurements, and finally the study's results.

### Quality assessment and bias

In this systematic review, we addressed missing data by implementing the following approach: (1) Data Extraction and Identification: During the data extraction process, we systematically identified and excluded articles with missing data that was essential for inclusion. (2) Reporting Transparency: We explicitly reported the missing data for each included study in our results sections. (3) Risk of Bias Assessment: Methodological quality, risk of bias, and study design were assessed in the current review by using the PEDro scale [48]. All of the included studies had a fair level of quality with a PEDro score of four to five [42–46,49,50], for an overview please refer to the descriptive summary of the seven included articles in Tables 1 and 2, and also for detailed review of the quality assessment of the included articles in S3. Finally, in the discussion section, we addressed the limitations imposed by missing data on our review's conclusions, and in the recommendations for future research.

A systematic review was carried out, following the principles and phases of the PRISMA model. A meta-analytical methodology was not appropriate due to the small number and heterogeneity of the selected articles.

## Results

The initial search results yielded 2814 citations. After removing duplicate results and initial screening of potentially relevant publications by titles and abstracts, 96 remained and were reviewed according to our inclusion/exclusion criteria. Eventually, seven publications were found eligible for this review (Fig 1).

### Inclusion/exclusion criteria set by the included studies

All participants were older than 18 years of age. Studies included only first-time unilateral CI users [42,43,45,49] who had the ability to understand English [46], mobility screening and history of hearing aid use for over three months [46]. Two studies did not mention the criteria [46,50].

Regarding exclusion criteria, studies excluded people with cognitive or physical impairments [46,49], psychological and anesthesia barriers [42], and individuals who failed to complete all scheduled assessments or follow the instructions [49,50]. Two studies did not mention any exclusion criteria [44,45].

### Sample size and basic characteristics

The sample size varied between studies, from one (out of three participants) [44] to 43 participants [43]. Overall, studies were balanced between sexes [43,44,46,49,50], except for one study that had more females than males in the control group [42]; one study did not describe the sex of participants [45]. The studies included CI users ranging in age from 19 to 84 years [42–44,46,49,50].

### Hearing status and assessments

The participants' hearing status varied in severity levels from normal hearing (for control groups) to severe hearing loss. One study used pure-tone audiometric testing before and after

**Table 1. Descriptive summary of included studies.**

| Author, Year | Study aims | Study design, settings, and population characteristics (N, age mean±sd) | Inclusion/Exclusion criteria | Hearing status pre/post intervention | Follow up | PEDro scale |
|---|---|---|---|---|---|---|
| Buhl et al. (2018) [49] | (1) Assess whether dynamic postural stability in CI recipients assessed by the functional gait assessment was below age-referenced norms before surgery and whether it decreased 4 weeks after surgery. (2) Evaluate whether there was a correlation between loss of residual hearing and a decrease in dynamic postural stability after cochlear implantation. | Prospective study Cochlear Implant (unilateral; most participants with similar CI types) N == 23, age 70±±15 | Inclusion: Age>>18; first-sided cochlear implantation (first time unilateral CI). Exclusion: Second-sided or bilateral cochlear implantation; inability to follow the instructions required for the FGA; orthopedic problems preventing normal gait; preexisting diagnosis associated with vertigo or a balance disturbance; high-grade vision impairment; alcohol abuse. | Pre: · mean pure-tone average [at 250 and 500 Hz] 64 dB HL (SD == 18 dB) Post: mean pure-tone 92 dB HL (SD == 21 dB); mean hearing loss after cochlear implantation was 28 dB (SD == 15 dB) | T1: preoperatively, the Mean duration between preoperative gait assessment and surgery was 2 (SD == 1.3) days. T2: Mean duration between surgery and the postoperative gait assessment was 37 (SD == 21) days. | 5 |
| Kaczmarczyk et al. (2019) [42] | Assess gait stability in patients before and after cochlear implantation (using two measures of gait stability: the B-coefficient and the gait stability ratio), and to propose a stability classification system based on the movement of the center of body mass (CoM) in the transverse plane. | Prospective study Study group: Cochlear Implant (unilateral) N==21, age 50.66±±18.02 Control group: N==30, age 45.6±±11.8 | Inclusion: adults identified as partial deafness patients, having normal low-frequency hearing but no hearing in the high-frequency range, scheduled for cochlear implantation (CI). Exclusion: Any significant pre-surgery psychological barriers; aversion to implantation; contraindications to receiving anesthesia during surgery. | Pre: First time unilateral CI- Partial deafness scheduled for cochlear implantation Post: · Normal hearing | T1: one day before surgery T2: three months after surgery with their cochlear implant turned off. The control group, was tested once | 5 |
| Kluenter et al. (2009) [43] | Estimate static and dynamic postural stability in severely hearing-impaired adult patients prior to and following CI surgery, and to compare the data with a group of healthy adults | Prospective study Study group: Cochlear Implant (unilateral) N==24, median age 51 (20-78) Control group: N==19, median age 49 (20-58) | Inclusion: age>>18; first time unilateral CI Exclusion: patients with bilateral implantation | Pre: Deafness, congenital hearing loss Post: None mentioned | T1: preoperatively, the median time between the first examination and surgery was 1 day and ranged from 1 to 320 days T2: Postoperatively, the median time between surgery and the second examination was 44 days, and ranged from 31 to 363 days | 5 |
| Le Nobel et al. (2016) [50] | Assess the effects of unilateral cochlear implantation (CI) on balance and the vestibular system in post-lingually deafened adults | Prospective study Cochlear Implant (unilateral) N==12, mean age 56 (20–78) | Not mentioned | Pre: patients with severe sensorineural hearing loss | T1: Preoperatively before surgery T2: Postoperatively, 1-week post-op T3: Postoperatively, 1-month post-op (CI Off). T4: Postoperatively, 1-month post-op (CI On) | 4 |
| Shayman et al. (2017) [44] | Evaluate whether wearing auditory assistive devices can improve gait and dynamic balance, testing with and without the use of their hearing assistive devices | Cross-sectional study Cochlear implant (bilateral): N == 1, age 21 Hearing aids (bilateral): N==2, age 38 and 82 | Not mentioned | Pre: Not mentioned for the patient with the cochlear implant; for hearing aids participants, HL of around 50 dB reported for first participants, HL around 80 dB at 8 kHz reported for second participant. | Not mentioned. All participants are experienced users. Testing was conducted with aided (on) condition and unaided (off) condition | 4 |

*(Continued)*

**Table 1.** (Continued)

| Author, Year | Study aims | Study design, settings, and population characteristics (N, age mean±sd) | Inclusion/Exclusion criteria | Hearing status pre/post intervention | Follow up | PEDro scale |
|---|---|---|---|---|---|---|
| Stieger et al. (2018) [45] | Study the medium effect (2 months postoperatively) of CI surgery on stance and gait balance control | Prospective study Cochlear Implant (unilateral) N == 30, age 59±±15.4 | Inclusion: First-time unilateral CI Exclusion: Not mentioned | Pre: patients with bilateral severe sensorineural hearing loss. Post: None mentioned | T1: Preoperatively 4 ±± 0.51 months before surgery T2: Postoperatively 2.1 ±± 0.87 after surgery == 1 month after the initial implant stimulation | 5 |
| Weaver et al. (2017) [46] | Assess if dynamic gait parameters change when hearing aids and cochlear implant (CI) processors are worn | Prospective study Cochlear Implant N == 12, mean age 55 (19–84) Mean years with bilateral aids: 5.4 (1-9 years) | Inclusion: age>>18; able to understand English; able to ambulate independently without a cane or walker; able to perform a Romberg test on a solid surface with eyes closed for 30 seconds; using a hearing aid device for over 3 months. Exclusion: cognitively impaired (Short-Blessed Test) | Pre: Unaided thresholds for all participants were worse than 30 dB in the better hearing ear. Post: All CI participants had severe to profound hearing loss bilaterally and could hear the testing stimulus while wearing their speech processors | T1: unaided condition (CI Off) T2: aided condition (CI On) T1 and T2 were conducted on the same day | 4 |

CI [49], one study used screening audiometric testing [46], and one study mentioned approximate hearing degree of hearing loss [44]. Specific auditory measures were not mentioned in four studies, but rather authors classified the degree of hearing loss as partial deafness, severe sensorineural hearing loss, and congenital hearing loss [42,43,45,50].

## Study design

It is important to note that the included studies used different methodologies. One approach was a prospective study that involved both pre-intervention and post-intervention assessments. The other approach assessed gait under on-off conditions, meaning gait was evaluated once with the CI turned on and once with it turned off. Five studies conducted pre- and post-intervention assessments [42,43,45,49,50], two studies assessed gait with the implant in both on and off conditions [44,46], and one study employed both pre- and post-implant assessments along with on-off evaluations [50].

## Timing of evaluation and follow-up

Pre-intervention and post-intervention assessments were conducted in five studies [42,43,45,49,50], of which four conducted a single post-CI evaluation [42,43,45,49], and one study conducted three post-CI evaluations [50]. The timing of follow-up varied between studies, both pre- and post- CI. Pre-CI baseline testing ranged from four months to one-day pre-CI [42,49]. One study had a range of one day to 320 days from baseline testing to CI [43], and one study did not mention the time of the pre-CI testing [50]. Similarly, inconsistency was seen in the timing of post-CI follow-up, with some studies conducted the tests for one to two months [45,49,50], and up to three months [42,46] post-CI. One study ranged from 31 to 363 days post-CI with a median of 44 days [42], one study did not mention the time of

**Table 2. Measures and main results presented in included studies.**

| Author, Year | Hearing measures | Vestibular measures | Main results |
|---|---|---|---|
| Buhl et al. (2018) [49] | Pure-tone audiograms performed prior to surgery and 4–6 weeks after surgery. | Video head impulse test (Otometrics, Natus Medical Denmark, Taarstrup, Denmark). Was tested only pre-operative and found normal in all subjects except two. | The difference between preoperative and postoperative gait assessment scores was not statistically significant after 5 weeks following surgery |
| Kaczmarczyk et al. (2019) [42] | None mentioned | Videonystagmography (VNG), vestibular evoked myogenic potentials (VEMP), the video head impulse test (vHIT), and computerized dynamic posturography sensory organization test (CDP-SOT) | In the intervention group, the gait stability ratio improved in 17 subjects after cochlear implantation, by an average of 6%; some specific parameters also showed statistically significant improvement between the two experimental group tests: step time ($p < 0.001$), single-support phase walking speed ($p < 0.05$), TW speed ($p < 0.001$) and CoM ($p < 0.05$) |
| Kluenter et al. (2009) [43] | None mentioned | Caloric testing | Walk Across Test: The step length and speed were significantly higher in the control group than in hearing-impaired patients before but also after CI surgery. The step width was not different.<br>The comparison before and after CI surgery showed that the mean speed during the walk decreased significantly. The other parameters did not change.<br>Tandem Walk: highly significantly worse in the patients than in the healthy adults. Only the mean end sway was not different between the control and the CI surgery group. The CI surgery did improve the mean step width but not the mean speed. Caloric testing on the operated side was unchanged in 20 patients, better in three patients, and worse in one patient. |
| Le Nobel et al. (2016) [50] | None mentioned | Patients' symptoms were assessed with questionnaires and the Dizziness Handicap Inventory (DHI). Subjective visual vertical (SVV) by using the Clear Health Media iPod application (Subjective Visual Vertical; Clear Health Media Inc., Wonga Park, Vic, Australia) | No statistically significant differences comparing pre- and post-operative TUG scores<br>No statistically significant difference was found with TUG scores implant on and off.<br>No statistically significant difference was found between pre- and postoperative DHI, and SVV testing<br>Head thrust testing did not demonstrate changes in the vestibular function in any of the operated ears between pre-operative and 1 month follow up assessments with the CI on or off. |
| Shayman et al. (2017) [44] | Approximate mention of hearing status | No vestibular tests | Gait velocity improved for each participant when wearing assistive device.Results on the mini-BEST test improved for all participants. The FAP score improved from the unaided (and off) to aided (on) auditory condition in all participants. Step length improved from the unaided (off) to aided (on) auditory condition in all participants. |
| Stieger et al. (2018) [45] | None mentioned | No vestibular tests | The overall pre- to postoperative trend indicated a not significant worsening of balance control as measured by the BCI.<br>By splitting up the patients into subgroups: (1) Those over 60 years of age and those with a balance deficit preoperatively had no significant worsening in balance control. (2) Significant worsening of balance during gait for those patients younger than 60 years of age at the time of implantation, and those with no balance pathology preoperatively. |
| Weaver et al. (2017) [46] | Screening audiometric testing | No vestibular tests | There was no statistically significant difference in gait outcomes between the aided (on) and unaided (off) conditions in both the HA and CI groups; There was no statistically significant difference in TUG times between aided (on) and unaided (off) conditions in both the HA and CI groups. |

evaluation [44], and one study conducted assessments with the processor on and off on the same day [46].

## Vestibular assessment

Three studies did not assess vestibular function [44–46]. The other four studies conducted various diagnostic and self-reported vestibular tests. Caloric tests were used in two studies [42,43], the Video Head Impulse Test (vHIT) was used in two studies [42,49], vestibular-Evoked Myogenic Potential (VEMP) was used in one study [42], one study used Head-Thrust

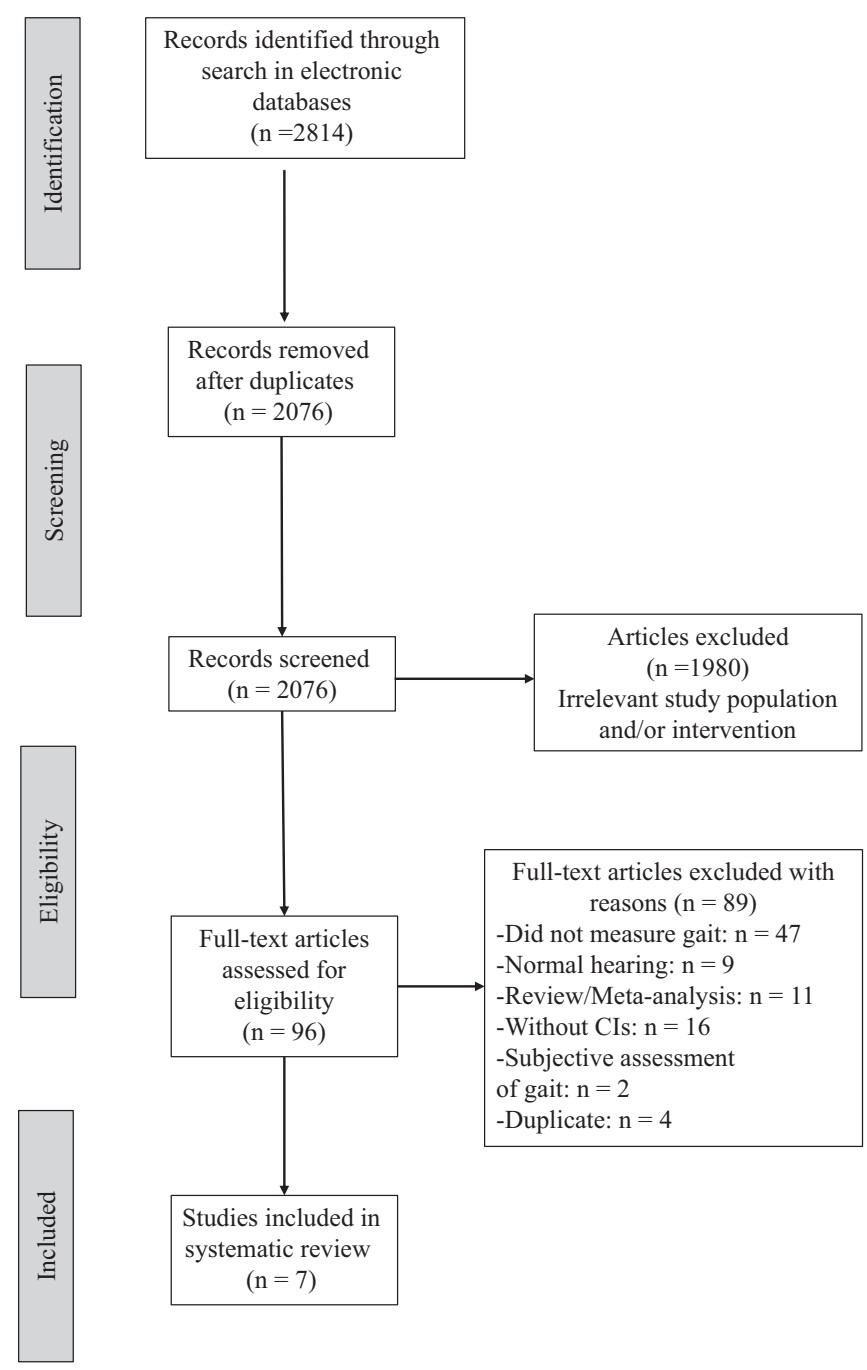

**Fig 1. The Preferred Reporting Items for Systematic Reviews and Meta-Analyses (PRISMA) flow diagram of the study identification, the screening, the eligibility, and the inclusion process within the systematic search.**

(HT) [50], and one used a self-report with the Dizziness Handicap Inventory (DHI) questionnaire together with the Subjective Visual Vertical (SVV) diagnostic test [50].

## Timing of vestibular assessment

Vestibular function was assessed only pre-operatively in one study and was found to be normal in all patients except two [49]. In another study, the vestibular assessment was conducted

pre-and post-operatively, but the results were not reported [42]. Two studies did not demonstrate a change between pre-operative and post-operative scores [43,50]. While the timeline for the vestibular assessment post-CI was not always specified, in one study, the postoperative median time between CI and the second examination was 44 days and ranged from 31 to 363 days [43], and in another study vestibular assessments were conducted postoperatively at one week and one month following CI [50]. Caloric testing on the operated side was unchanged in 20 patients, better in three patients, and worse in one patient [43], no statistically significant difference was found between pre-operative and postoperative DHI, and SVV testing, and head thrust testing did not demonstrate changes in the vestibular function in any of the operated ears between pre-operative and post-operative assessments with the CI on or off [50], the vHIT was used only pre-operatively in [49].

### Etiology

Participants in two studies had bilateral sensorineural hearing loss [45,46], while other studies did not mention whether participants had unilateral or bilateral hearing loss. The etiology of the hearing loss among most participants across studies was mostly idiopathic. Nevertheless, three studies mentioned additional etiology with a mixed sample: one mentioned syndromic hearing loss or infectious deafness [43], another study mentioned noise-induced, autoimmune diseases, sudden hearing loss, measles, ototoxic drug exposure, mitochondrial disease, and congenital hearing loss [50], and the third study declared a history of Meniere's disease, history of congenital cytomegalovirus infection, and a history of benign positional vertigo with normal results on the caloric test [44].

### Hearing Intervention

Five of the studies included participants who underwent unilateral CI [42,43,45,49,50], and two studies included bilateral CIs [44,46].

### Gait assessment

All studies measured gait performance on various walking tasks. These tasks and associated gait performance metrics are detailed in Table 3. Overall, the following gait tasks were performed: The Time Up and GO (TUG), Walk Across (WA), Tandem Walk (TW), Functional Gait Assessment (FGA), Gait Stability Ratio (GSR), Center of Mass (CoM) determination, Mini BEST Test, and Functional Ambulation Performance (FAP). Overall, the following metrics were recorded: Gait speed, stride length, gait variability, step width, straightness, single support, double support, cadence, and stability of gait [42–44,46].

### Gait outcomes following CI

Gait outcomes following CI varied between improvement, worsening, and no change from baseline to follow-up. These findings varied by the outcome measure, the task tested, duration of follow-up, and on-off conditions.

Among the studies that reported *improvement in gait*: One study showed that a small subset of participants improved on outcome measures (gait velocity, stride length variability, swing time variability, and the double support phase) after more than three months post-implant activation, despite no difference between on-off conditions [46]. Another study found improved gait velocity (timing of evaluation not mentioned) when implants were turned on (compared to off) [44]. Another study showed improvement in GSR in 17 out of 21 patients with three months of follow-up. This study also reported statistically significant improvement

**Table 3. Gait assessment, description, instruments, and variables.**

|  | **Test and description** | **Instruments for gait analysis** |
|---|---|---|
| Buhl et al. (2018) [49] | Functional Gait Assessment (FGA): Consists of ten walking tests on a six-meter surface, including regular walking, pace change during walking, and a 3.6-meter tandem walk | Each test was rated from zero to three and combined for a total score. The instrument was not specified |
| Kaczmarczyk et al. (2019) [42] | Walking 10 meters at a preferred speed | Gait speed, cadence, stride and step time, stride and step length, step width, the timing of single support and double support were collected with the Motion Capture System (Vicon Motion Systems Ltd, Oxford, UK). Gait Stability Ratio (GSR): Was calculated based on the cadence and velocity measures using the equation GSR = cadence/velocity. Center of Mass (CoM): Evaluating gait stability in terms of the angular deviation of the CoM trajectory and straightness of gait |
| Kluenter et al. (2009) [43] | Walk Across (WA): Subjects walked from one end of a force plate to the other over three trials (length was not mentioned) Tandem Walk (TW): Subjects walked a 'tightrope' from heel to toe over three trials (length was not mentioned) | For WA - Step width, step length, gait speed, and step length symmetry were collected with a Balance Master computerized force plate (NeuroCom, Clackamas, OR, USA). For TW - Step width, gait mean speed, and the mean end sway (the mean anterior/posterior sway for the first 5 seconds after stopping the task) were collected with a Balance Master computerized force plate (NeuroCom, Clackamas, OR, USA) |
| Le Nobel et al. (2016) [50] | Timed Up and GO (TUG): Measures the time required for a participant to rise from a sitting position, walk ten feet, and return | The time taken to complete the TUG test was recorded |
| Shayman et al. (2017) [44] | Mini-Balance Evaluation Systems Test (Mini-BESTest): Walking on a seven-meter strip and TUG | Velocity, step length, dynamic base of support, and step symmetry ratio were collected with a pressure sensor system (GAITRite, CIR Systems, Sparta, NJ) |
| Stieger et al. (2018) [45] | Tandem Walk (TW): Subjects walked a 'tightrope' from heel to toe over three trials (length was not mentioned) Walking task: Subjects walked three meters with eyes closed, walked three meters while pitching the head up and down, and walked three meters while rotating the head left and right | Roll angle, pitch angle, roll angular velocity, pitch velocity, and task duration were collected with the SwayStarTM device (Balance International Innovations GmbH, Switzerland) |
| Weaver et al. (2017) [46] | Timed Up and GO (TUG): Measures the time required for a participant to rise from a sitting position, walk ten feet, and return Gait task: Participants were blindfolded and instructed to walk at their normal comfortable pace, straight ahead to a source of sound, and to use the sound to help orient and balance if desired | For TUG - Time taken to complete the TUG test was recorded Gait characteristics, including gait velocity, stride length variability, swing time variability, time in double support phase, cadence, and stride length, were collected with Opal inertial sensors (APDM, Portland, OR) |

in several other parameters, such as step time, single support phase walking speed, TW, speed, and CoM determination [42].

Among the studies that reported *worsening or no change in gait*, one study reported a significant worsening in walking tasks one month after CI, in patients that were younger than 60 at the time of CI without a balance pathology. Other participants showed no change [45]. Another study did not find a statistically significant difference in TUG performance one-month post-CI [50]. Similarly, five weeks post-CI, there was no significant difference in FGA scores [49]. Lastly, Kluenter et al. [43] found a significant decrease in the mean walking speed in the WA task, but no change in other parameters. While in the TW task, the mean step

width improved, the mean speed did not increase after the CI (median time postoperatively was 44 days and ranged from 31 to 363 days) [43].

In addition, it is important to highlight the improved outcomes of gait based on the *methodological approaches* introduced above that utilized prospective design or on-off conditions. By utilizing prospective methods, one study [42] indicated an improvement in gait performance three months post-implant gait stability ratio, step time, single-support phase, tandem walk speed, and CoM. In contrast, studies with follow-ups averaging between 37 days and two months reported either a decline or no change in outcomes after CI (43, 45, 49). Studies using the on-off design reported mixed outcomes. Two studies reported no difference between the implant being turned on and off (46, 50), while another study noted improved gait velocity, when implants were activated compared to the off state [44].

## Discussion

This systematic review synthesized seven studies that examined the effect of hearing interventions by CI on gait performance among adults with hearing loss. A three-month follow-up period post-CI showed improvement in some gait measurements, such as Tandem Walk speed, but not in walking speed. Also, the seven included studies used several different gait tasks, alone or in combination, to assess gait performance with different measurement methods and devices.

Importantly, some methodological gaps were identified. One gap refers to insufficient periods of post-implant follow-up, and another gap refers to non-uniform measurements and tasks used to quantify gait, making it difficult to compare between studies and impossible to conduct a meaningful meta-analysis. In addition, studies lacked assessment of underlying behavioral and neural mechanisms such as cognitive tests, neural monitoring, or dual-task testing.

### Follow-up periods

When patients were evaluated between four to six weeks post-CI, they showed worse gait outcomes [45] or no improvement [43,49,50], possibly due to acute symptoms related to the surgical procedure and activation of the implant. However, when tested three months post-CI, patients showed improvement in some gait parameters, such as tandem walk speed, mean step width, and step time, but not in comfortable normal walking speed [42,44]. This is consistent with studies that evaluated static and dynamic balance outcomes for longer follow-up. These studies showed a consistent improvement in postural stability over a duration of one year [51] and two years post-CI [52]. Additional studies with longer follow-up (e.g., one year) are required for assessing gait and any associated neuroplastic brain changes associated with CI [32].

### Assessment of underlying behavioral and neural mechanisms

Gait outcomes changed between pre- to post-CI conditions, yet none of the studies in this review looked into possible underlying mechanisms (sensory, physiological, cognitive, behavioral), which limited our understanding of the mechanisms underlying the association between hearing and gait and how CI affects gait.

The studies in this review did not conduct any neural monitoring to assess neural activity and evaluate possible neuroplastic changes after the CI. Indeed, monitoring neural activity may demonstrate neural plasticity and compensatory activity in line with the Scaffolding Theory of Aging and Cognition (STAC) that suggests there may be qualitative differences in neural activation and behavioral strategies [53], or show changes in neural patterns when one domain becomes challenged [53]. In such cases, neural monitoring using real-time functional neuroimaging [53] tools can demonstrate the changes in neural activity patterns between pre- and post-CI.

Neural deprivation induced by reduced audio-sensory input can alter the central auditory system and modify the organization of the cortex through cross-modal re-organization [27]. This re-organization results in compensation by other neural systems to maximize performance; that is, using other brain structures or networks to increase the activity and compensatory expansion to other modalities [18]. Nevertheless, the decline is not permanent and can respond to changes in acoustic experience and cortical function can improve [28,29], through the use of hearing aids in mild-to-moderate hearing impairments, and may reverse some of the central changes related to severe-to-profound hearing loss by restoring afferent activity through CI [30]. Functional neuromonitoring contributes to the assessment and evaluation of neural activity and to identifying the brain systems responsible for different behaviors [54]. Thus, neural activity monitoring uniquely contributes to understanding neural activity during gait [55], compatible with CI devices, not subject to electrical artifacts and flexible in auditory stimuli paradigms [56].

## Dual task of gait and cognition

In addition, and as introduced above, cognitive function can improve through the use of hearing aids in cases of mild-to-moderate hearing impairments. This change is often observed six months from the fitting to accommodate the neuroplastic changes [27,28]. None of the studies included in this review mentioned any cognitive evaluations. This evaluation is important because gait functioning is linked to cognitive performance [57,58], and cognitive demands increase as sensory input decreases [59], resulting in reduced walking speed, step length [60], and gait asymmetry [61].

The role of cognition in gait performance can be assessed with dual-task evaluations that combine cognitive tasks with walking after a CI [62]. Cognitive demands are likely to be higher when another cognitive task accompanies walking (i.e., dual tasking), also in challenged neural conditions, when walking becomes more cognitively demanding and reflective of real-life walking, leading to an increased risk of falling [57,63]. For individuals with hearing loss, conducting a dual-task leads to prioritizing the maintenance of posture over cognitive performance [19] by using alternative cognitive resources [18] resulting in changes in the neural system [64]. Hence, gait monitoring under dual-task conditions, i.e., walking with a cognitive task, after CI, should be of interest among researchers, yet these tasks were omitted in the studies included in this review.

## Vestibular function post-CI

After reviewing the current data, it was found that there was a large variability in the choice and timeline of vestibular testing. The vestibular system and the cochlea are located in close proximity within the inner ear [2,16], and the literature, in general, is equivocal about the connection between vestibular function and CI. On one hand, vestibular function worsened after a CI in caloric testing as well as in VEMP, and no significant post-CI effect was found in HIT, posturography and DHI scores among individuals who underwent a CI [65]. On the other hand, others show improvement in performance on dynamic platform posturography [52]. Possible explanations for worsening in vestibular function, may include trauma to the cochlea due to the surgical procedure [65], infections due to anatomical proximity between the vestibular organs and the cochlea [2,16], and the timing of post-CI testing.

## Consistency in measurements

This review revealed a lack of consistency in the tasks and outcomes used to assess gait performance. The seven included studies used seven different gait tasks, alone or in combination, to

assess gait performance and no two studies' methods were identical, with different measurement methods and devices. This inconsistency in the gait measurements makes it challenging to compare the results between studies. A more standardized approach to gait assessment using sensor-based gait evaluations could increase ecological assessment and diagnostic accuracy [66,67], and could lead to floor and ceiling effects [68]. Cognitive evaluations and dual-task testing are also needed to improve the comparability of findings and advance our understanding of the impact of CIs on gait performance among individuals with hearing loss.

## Study design and implementation of on-off methods

This review highlights two key study designs: Prospective [42,43,45,49,50] and on-off designs, with hearing input is assessed with the CI turned on (on condition) and off (off condition) [44,46]. One of these studies [50] employed a mixed-method approach, utilizing both prospective and on-off designs. In the prospective approach, evaluations were conducted before and after implantation to determine whether CIs improve gait. The review identified variations in follow-up duration and outcomes among these studies, with longer follow-up periods (up to three months) showing gait improvements. In contrast, the on-off approach did not demonstrate gait improvement when comparing CI-on and CI-off conditions. However, this approach may enhance our understanding of the mechanisms underlying post-implant gait improvements, which are thought to arise from neuroplastic changes in response to increased audibility [29]. Combining these two approaches could provide a more comprehensive understanding of gait improvements induced by cochlear implantation and audibility enhancement. Nonetheless, longer durations of CI use may be necessary to observe significant and sustained outcomes.

## Sample size

The sample size varied widely across the studies, with some having as few as one participant [44], and others including up to 51 individuals [42]. This limited number of participants may not provide sufficient statistical power to detect meaningful changes in gait performance following CI. Additionally, a small sample size can lead to reduced generalizability of the findings. A larger participant pool would better represent the population of adults with hearing loss and improve the robustness of the conclusions drawn from the studies.

## Hearing status

The systematic review identified the variability in the participants' hearing statuses as a significant gap in the existing literature. The reviewed studies included individuals with various types of hearing loss, as well as normal-hearing control groups for comparison. Some studies did not provide specific auditory measures, while others mentioned participants having bilateral severe to profound sensorineural hearing loss or partial deafness. This variability in the participants' hearing statuses makes it challenging to draw generalizable conclusions about the effect of CIs on gait performance among adults with hearing loss. A more standardized approach for reporting audiometric measures and separating unilateral from bilateral hearing loss would enhance the consistency and comparability of future studies in this area.

## Limitations

This review is limited by the small number of studies identified (seven), the small sample size per study, and the 'fair' quality of the research as per the PEDro scale. In addition, the follow-up period in the studies was, in most cases, insufficient to accommodate the neuroplastic changes

following CI. Furthermore, inconsistency was observed in gait tasks, hearing status, and vestibular testing. Given those constraints, a meta-analysis was not possible.

Our knowledge of the relationship between hearing and gait and the mechanisms underlying it is constrained by the absence of cognitive assessments and neural monitoring. The studies that were reviewed also had limitations despite meeting inclusion criteria. These limitations included a small number of participants, the absence of real-world daily activities in actual conditions, specific interventions such as single-sided CIs, and the lack of randomized controlled trials.

While not the topic of the current review, during our thorough search of available literature, we observed a significant gap in research regarding the impact of hearing aids on the gait of hearing aid users. We only came across one longitudinal study [10] that investigated the relationship between gait and hearing aids, and two studies that examined both CI and hearing aid users [44,46]. This limited number of studies precludes synthesis and conclusions.

## Conclusion

To our knowledge, this is the first review to investigate the effect of CI on gait performance, as previous reviews have investigated the effect of hearing aids, or hearing aids and CIs, on static and dynamic balance.

We found that short follow-up (up to two months) after the activation of the CI was associated with unchanged or even worse gait quality, and a longer follow-up (at least three months) was associated with a partially improved gait such as in mean step width, step time, and Tandem Walk speed but not in comfortable, normal walking speed.

## Recommendation for future research

Future studies should assess cognition and monitor neural activity during gait under dual-task conditions to further understand the underlying mechanisms between hearing and gait. In addition, randomized controlled trials with long-term follow-ups are needed to determine the effect of CI on gait function and identify the optimal follow-up periods. The implementation of on-off evaluations should also be considered to better understand the mechanistic relationship between hearing and gait. Combining prospective and on-off designs, with post-implant on-off evaluations conducted beyond three months after implantation, could reveal significant outcomes and provide deeper insights into the mechanisms underlying post-CI improvements.

Finally, the field will benefit from consistency [50] in the gait assessment protocol and outcome measures using challenging tasks with instrumented assessments [65].

## Recommendation for clinical practice

Clinically, it is crucial to assess the vestibular and physical function, fall risk, and cognitive abilities of patients with hearing loss prior to undergoing a CI procedure. Also, efforts must be made to follow a strict protocol with tasks, baseline measures, and longer follow-ups. This will assist in personalizing the rehabilitation program after the implantation.

## Supporting information

**S1. The PRISMA 2020 statement.**
(DOCX)

**S2. Description of 2076 records.**
(XLSX)

**S3. Supporting information for quality assessment.**
(DOCX)

## Author contributions

**Conceptualization:** Bahaa Rafoul, Hanin Karawani, Maayan Agmon.

**Formal analysis:** Bahaa Rafoul.

**Funding acquisition:** Mauricio Cohen-Vaizer, Hanin Karawani, Maayan Agmon.

**Investigation:** Bahaa Rafoul, Roy Tzemah-Shahar.

**Methodology:** Bahaa Rafoul, Hanin Karawani, Maayan Agmon.

**Resources:** Hanin Karawani, Maayan Agmon.

**Supervision:** Hanin Karawani, Maayan Agmon.

**Writing – original draft:** Bahaa Rafoul.

**Writing – review & editing:** Anat V. Lubetzky, Mauricio Cohen-Vaizer, Hanin Karawani, Maayan Agmon.

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
