## [Decision Letter · Decision Letter 0]

11 Nov 2024

PONE-D-24-44082Effects of cochlear implantation on gait performance in adults with hearing impairment: A systematic reviewPLOS ONE

Dear Dr. Karawani,

Thank you for submitting your manuscript to PLOS ONE. After careful consideration, we feel that it has merit but does not fully meet PLOS ONE’s publication criteria as it currently stands. Therefore, we invite you to submit a revised version of the manuscript that addresses the points raised during the review process.

We look forward to receiving your revised manuscript.

Kind regards,

Renato S. Melo, PhD

Academic Editor

PLOS ONE

“This work was partially supported by the Faculty of Social Welfare and Health Sciences grant (awarded to HK and MA], Rambam and Faculty of Social Welfare and Health Sciences dual grant [awarded to BR, MCV, HK, MA]; The MAOF Research Grant Program for Research in Nursing at the Rambam health care campus [awarded to BR].”

“This work was supported by the Faculty of Social Welfare and Health Sciences grant (awarded to HK and MA], Rambam and Faculty of Social Welfare and Health Sciences dual grant [awarded to BR, MCV, HK, MA]; The MAOF Research Grant Program for Research in Nursing at the Rambam health care campus [awarded to BR].

The authors have no relevant financial or non-financial interests to disclose.”

“This work was partially supported by the Faculty of Social Welfare and Health Sciences grant (awarded to HK and MA], Rambam and Faculty of Social Welfare and Health Sciences dual grant [awarded to BR, MCV, HK, MA]; The MAOF Research Grant Program for Research in Nursing at the Rambam health care campus [awarded to BR].”

5. As required by our policy on Data Availability, please ensure your manuscript or supplementary information includes the following:

Additional Editor Comments:

After careful review, the reviewers believe that the manuscript has the potential to be accepted for publication, however, the manuscript still needs adjustments before this can occur.

Reviewers' comments:

Reviewer's Responses to Questions

**Comments to the Author**

1. Is the manuscript technically sound, and do the data support the conclusions?

Reviewer #1: Yes

Reviewer #2: Yes

2. Has the statistical analysis been performed appropriately and rigorously? 

Reviewer #1: N/A

Reviewer #2: Yes

3. Have the authors made all data underlying the findings in their manuscript fully available?

Reviewer #1: Yes

Reviewer #2: Yes

4. Is the manuscript presented in an intelligible fashion and written in standard English?

Reviewer #1: Yes

Reviewer #2: Yes

5. Review Comments to the Author

Reviewer #1: The studies in this systematic review address two separate research questions:

1) Does cochlear implantation improve gait? This can only be addressed by the prospective studies that measured gait before and after implantation. There are 5 such studies: Buhl (2018), Kaczmarczyk (2019), Kluenter (2009), Le Nobel (2016), Stieger (2018).

2) If you have a CI, is gait affected by whether the CI is on or off? This question is addressed by 3 studies: Shayman (2017), Weaver (2017), and also Le Nobel (2016) (which addresses both research questions).

I would like the authors of this systemic review to more explicitly distinguish between these two research questions, and summarise the corresponding evidence separately.

Research question 1 can inform the important clinical decision of whether or not to implant; CI surgery is expected to degrade residual acoustic hearing, and may have a negative impact on vestibular function.

Regarding research question 2, comparisons of gait with CI off and on may improve the understanding of the mechanisms behind the observed post-op gait improvement relative to pre-op.

A minor point is that although the authors here say that Shayman (2017) had N = 3, only one participant had cochlear implants, so it is actually N = 1 for the purpose of this systematic review.

Reviewer #2: Thank you for reviewing this study, which presents an interesting topic for the field of audiology and motor control.

I congratulate the authors on the study; however, I felt that a more robust and evidence-based justification was needed as to why these individuals' gait improved after CI.

I suggest that the authors write one or two paragraphs showing that the same thing happens in children with hearing loss who underwent CI surgery. The evidence shows that hearing again improves these individuals' balance, and with better balance, other motor skills that depend on balance to be performed, such as walking, can improve, both in deaf children with lesser degrees of hearing loss and in those who use CI. Below are the suggested references for the authors to write the requested paragraphs.

Cushing SL, Pothier D, Hughes C, Hubbard BJ, Gordon KA, Papsin BC. Providing auditory cues to improve stability in children who are deaf. Laryngoscope. 2012; 122: 101-2. Doi: 10.1002/lary.23807.

De Kegel A, Maes L, Baetens T, Dhooge I, Van Waelvelde H. The influence of a vestibular dysfunction on the motor development of hearing-impaired children. Laryngoscope. 2012; 122: 2837-43. Doi: 10.1002/lary.23529.

Hamzehpour F, Absalan A, Pirasteh E, Sharafi Z, Arbabsarjoo H. Investigating the effect of hearing aid use on the balance status of children with severe to profound congenital hearing loss using the pediatric clinical test of sensory interaction for balance. J Am Acad Audiol. 2021; 32: 303-7. Doi: 10.1055/s-0041-1728754.

Mazaheryazdi M, Moossavi A, Sarrafzadah J, Talebian S, Jalaie S. Study of the effects of hearing on static and dynamic postural function in children using cochlear implants. Int J Pediatr Otorhinolaryngol. 2017; 100: 18-22. Doi: 10.1016/j.ijporl.2017.06.002.

Melo RS, Lemos A, Macky CFST, Raposo MCF, Ferraz KM. Postural control assessment in students with normal hearing and sensorineural hearing loss. Braz J Otorhinolaryngol. 2015; 81: 431-8. Doi: 10.1016/j.bjorl.2014.08.014. Doi: 10.1016/j.bjorl.2014.08.014.

Melo RS, Lemos A, Raposo MCF, Belian RB, Ferraz KM. Balance performance of children and adolescents with sensorineural hearing loss: Repercussions of hearing loss degrees and etiological factors. Int J Pediatr Otorhinolaryngol. 2018; 110: 16-21. Doi: 10.1016/j.ijporl.2018.04.016.

Melo RS, Lemos A, Raposo MCF, Monteiro MG, Lambertz D, Ferraz KM. Repercussions of the hearing loss degrees and vestibular dysfunction on the static balance of children with sensorineural hearing loss. Phys Ther. 2021; 101: pzab177. Doi: 10.1093/ptj/pzab177.

Melo RS, Lemos A, Wiesiolek CC, Soares LGM, Raposo MCF, Lambertz D, et al. Postural sway velocity of deaf children with and without vestibular dysfunction. Sensors (Basel). 2024; 24: 3888. Doi: 10.3390/s24123888.

Suarez H, Alonso R, Arocena S, Ferreira E, Roman CS, Suarez A, et al. Sensorimotor interaction in deaf children. Relationship between gait performance and hearing input during childhood assessed in pre-lingual cochlear implant users. Acta Otolaryngol. 2017; 137: 346-51. Doi: 10.1080/00016489.2016.1247496.

Wolter NE, Gordon KA, Campos J, Madrigal LDV, Papsin BC, Cushing SL. Impact of the sensory environment on balance in children with bilateral cochleovestibular loss. Hear Res. 2021; 400: 108134. Doi: 10.1016/j.heares.2020.108134.

Zarei H, Norasteh AA, King L. The effect of auditory cues on static postural control: a systematic review and meta-analysis. Audiol Neurotol. 2022; 27: 427-36. Doi: 10.1159/000525951.

6. PLOS authors have the option to publish the peer review history of their article (what does this mean? ). If published, this will include your full peer review and any attached files.

**Do you want your identity to be public for this peer review?** For information about this choice, including consent withdrawal, please see our Privacy Policy .

Reviewer #1: **Yes: ** Brett Swanson

Reviewer #2: No

---

## [Author Response · Author response to Decision Letter 1]

14 Jan 2025

We have uploaded a Response letter that addresses each comment point by point

---

## [Decision Letter · Decision Letter 1]

31 Jan 2025

Effects of cochlear implantation on gait performance in adults with hearing impairment: A systematic review

PONE-D-24-44082R1

Dear Dr. Karawani,

We’re pleased to inform you that your manuscript has been judged scientifically suitable for publication and will be formally accepted for publication once it meets all outstanding technical requirements.

Kind regards,

Renato S. Melo, PhD

Academic Editor

PLOS ONE

Additional Editor Comments (optional):

Reviewers' comments:

Reviewer's Responses to Questions

**Comments to the Author**

1. If the authors have adequately addressed your comments raised in a previous round of review and you feel that this manuscript is now acceptable for publication, you may indicate that here to bypass the “Comments to the Author” section, enter your conflict of interest statement in the “Confidential to Editor” section, and submit your "Accept" recommendation.

Reviewer #1: All comments have been addressed

Reviewer #2: All comments have been addressed

2. Is the manuscript technically sound, and do the data support the conclusions?

Reviewer #1: Yes

Reviewer #2: Yes

3. Has the statistical analysis been performed appropriately and rigorously? 

Reviewer #1: N/A

Reviewer #2: Yes

4. Have the authors made all data underlying the findings in their manuscript fully available?

Reviewer #1: Yes

Reviewer #2: Yes

5. Is the manuscript presented in an intelligible fashion and written in standard English?

Reviewer #1: Yes

Reviewer #2: Yes

6. Review Comments to the Author

Reviewer #1: (No Response)

Reviewer #2: I congratulate the authors for the changes, all my requests were met by the authors in the manuscript, which I believe is suitable for publication.

7. PLOS authors have the option to publish the peer review history of their article (what does this mean? ). If published, this will include your full peer review and any attached files.

**Do you want your identity to be public for this peer review?** For information about this choice, including consent withdrawal, please see our Privacy Policy .

Reviewer #1: **Yes: ** Brett Swanson

Reviewer #2: No

---

## [Editor Report · Acceptance letter]

PONE-D-24-44082R1

PLOS ONE

Dear Dr. Karawani,

I'm pleased to inform you that your manuscript has been deemed suitable for publication in PLOS ONE. Congratulations! Your manuscript is now being handed over to our production team.

Kind regards,

on behalf of

Dr. Renato S. Melo

Academic Editor

PLOS ONE